# Desmoid Tumors in the Active Surveillance Era: Evaluation of Treatment Options and Pain Relief in a Single-Center Retrospective Analysis

**DOI:** 10.3390/jpm13121653

**Published:** 2023-11-27

**Authors:** Marco Rastrelli, Benedetta Chiusole, Francesco Cavallin, Paolo Del Fiore, Andrea Angelini, Maria Chiara Cerchiaro, Pietro Ruggieri, Marta Sbaraglia, Simone Mocellin, Antonella Brunello

**Affiliations:** 1Soft-Tissue, Peritoneum and Melanoma Surgical Oncology Unit, Veneto Institute of Oncology IOV—IRCCS, 35128 Padua, Italy; 2Department of Surgery, Oncology and Gastroenterology (DISCOG), University of Padua, 35128 Padua, Italy; 3Oncology 1, Department of Oncology, Veneto Institute of Oncology IOV—IRCCS, 35128 Padua, Italy; 4Independent Statistician, 36020 Solagna, Italy; 5Department of Orthopedics and Orthopedic Oncology, University of Padova, 35128 Padua, Italypietro.ruggieri@unipd.it (P.R.); 6Department of Medicine (DIMED), University of Padua School of Medicine, 35128 Padua, Italy; 7Department of Pathology, Azienda Ospedale Università Padova, 35128 Padua, Italy

**Keywords:** desmoid tumor, active surveillance, therapeutic management

## Abstract

In patients with desmoid tumors (DTs), active surveillance has been increasingly preferred over surgery, while treatment (including pharmacological therapy, radiotherapy, and/or surgery) is performed in cases with confirmed disease progression. This study aimed to evaluate event-free survival and pain management according to different treatment strategies. We evaluated event-free survival, including recurrence after initial surgical treatment or changes in the therapeutic management after initial non-surgical treatment and pain management according to different treatment strategies. All patients referred for DT in 2001–2021 at our institutions were stratified into four groups: those treated surgically prior to 2012 (SGPre12) or after 2012 (SGPost12), those treated pharmacologically (MG), and those under active surveillance (ASG). An event was defined as recurrence after initial surgical treatment or a change in therapeutic management. Overall, 123 patients were included in the study: 28 in SGPre12, 41 in SGPost12, 38 in MG, and 16 in ASG. Pharmacological treatment resolved painful symptoms in 16/27 (60%) patients (*p* = 0.0001). The median follow-up duration was 40 months (IQR 23–74). Event-free survival at 1, 3, and 5 years was: 85%, 70%, and 62% in SGPre12; 76%, 58%, and 49% in SGPost12; 49%, 31%, and 31% in MG; and 45%, 45%, and 45% in ASG. Our findings support the role of active surveillance as initial management, as demonstrated by the fact that about half the patients did not experience any progression, while surgery can be reserved as a first-line approach for selected patients. In terms of pain relief, medical therapy led to symptom resolution in more than half the cases.

## 1. Introduction

Desmoid tumors (DTs), otherwise known as desmoid fibromatosis or aggressive fibromatosis, are a locally aggressive, non-metastasizing mesenchymal tumor with an invasive growth pattern, no metastatic potential, and a high propensity for locoregional recurrence after surgery. These aspects result from the clonal proliferation of fibroblasts, connective tissue cells responsible for tissue support, which play a role in wound healing. DNA mutations in these cells cause their uncontrolled growth, leading to the formation of DT. These tumors are characterized by infiltrative growth and a tendency towards local recurrence, which is why they are also referred to as ‘aggregative fibromatosis’, but they do not have the potential to metastasize, i.e., the ability to spread to other tissues. However, DTs can sometimes be multifocal.

Desmoid tumors account for less than 3% of soft tissue tumors, frequently affect women of childbearing age, and are linked to pregnancy in a subset of cases [1]. Most cases are sporadic and, in these cases, mutations in the beta-catenin gene (CTNNB1) are usually found [1]. About 5–10% of cases occur in patients with familial adenomatous polyposis (FAP), due to an APC (adenomatous polyposis coli) germ line mutation, with the two mutations being mutually exclusive [2]. Patients suffering from these heredo-familial syndromes usually discover that they have DTs during routine examinations performed for screening purposes. DTs can occur in both somatic and visceral sites. Extra-abdominal, abdominal wall, and intra-abdominal are common locations [1], and different prognoses have been associated with different locations [3].

Sporadic DTs are so called because the cause is not known. In 85% of cases, there are somatic mutations of the CTNNB1 gene (3q21), the gene coding for beta-catenin (i.e., mutations occurring in a single cell that transmits to its daughter cells, but the mutation is not inherited by the patient’s offspring). In this case, there is only the presence of the desmoid tumor, without the simultaneous presence of other clinical manifestations, as is the case with FAP. DTs in FAP patients account for only 10% of all cases. In these cases, the mutation is at the germ line level and concerns the APC oncosuppressor gene, which is responsible for the control of cell growth and death and present on chromosome 5 (5q21–q22), which cod-codes for the colon adenomatous polyposis protein, a protein that is, therefore, mutated in all the patient’s cells. For this reason, the syndrome can be transmitted.

Beta-catenin and APC mutations appear to be mutually exclusive, so the identification of a somatic beta-catenin mutation can help rule out a systemic condition, such as FAP, in the patient. Thus, the patient will have either a sporadic desmoid or the APC mutation typical of FAP, as it is not possible to have both. Conversely, the wild-type (lack of mutation) beta-catenin status should raise FAP suspicion; in this case, it is recommended to study family history for FAP and/or perform a colonoscopy to exclude it.

In people without FAP, DTs generally manifest as masses of varying size with a hard-ligneous consistency, which, however, can only be appreciated if it arises in superficial areas. Often physicians who are not specialized in this pathology, given its rarity, mistake the initial diagnosis, confusing it with a low-grade lipoma or liposarcoma, and go on to surgically remove it before a biopsy has been taken. Surgery can be charged with a high risk of recurrence, regardless of the adequacy of the margins. For this reason, it is important to refer patients to a specialist in sarcoma surgery/oncology at one of the referral centers before making any decisions.

The initial diagnosis is based on imaging techniques (computed tomography and magnetic resonance imaging), which can reveal the presence of an expanding, infiltrating mass. Nuclear magnetic resonance imaging (MRI) is the method of choice for DTs, except for intra-abdominal DTs, where CT scans are preferred, at least at the early diagnostic stage. MRI reveals the presence of an expanding, infiltrating mass with unclear margins and diffuse contrast medium uptake (enhancement). The diagnostic key in the MRI of a desmoid tumor lies in the hypointense bands identifiable at T2, and an association has been shown between growth and high T2 signal intensity.

The diagnosis is confirmed by biopsy of the tumor, which shows abundant collagen around elongated fusiform cells containing small, regular nuclei and pale cytoplasm.

Macroscopically, DTs present as gray–brownish neoformations, with a fasciculated appearance when cut, a tense, elastic consistency, translucent appearance, and in continuity with the muscle tissue; this characteristic is also evident microscopically, with infiltrative characteristics towards striated muscle tissue. This characteristic lack of a boundary with the healthy tissue explains the high local recurrence and the great intraoperative difficulty in macroscopically defining the limits of the resection.

Immunohistological examination shows the expression of muscle cell markers (actin, desmin, vimentin) and the absence of CD34. In addition, the diagnosis can be confirmed by screening for CTNNB1 mutations.

Differential diagnosis is wide-ranging and includes, on the one hand, fibrosarcomas and, on the other, myofibroblastic processes, such as nodular fasciitis, hypertrophic scars, and keloids. The differential diagnosis of intra-abdominal DT arises with gastrointestinal stromal tumors, isolated fibrous tumors, inflammatory myofibroblastic tumors, sclerosing mesenteritis, and retroperitoneal fibrosis.

Historically, surgery has been the mainstay of treatment and requires the removal of the tumor with a wide margin, i.e., it is associated with the removal of a large area of healthy tissue. Despite the adequacy of the margins, the risk of locoregional recurrence and the morbidity associated with repeated surgery encourages the search for new therapeutic strategies [3,4]. Currently, a “wait-and-see” approach based on active surveillance is preferred, especially for asymptomatic patients, and treatment, including pharmacological therapy, radiotherapy, and/or surgery, should be postponed until the disease has demonstrated progression [5,6,7]. Non-invasive treatments are generally preferred over surgery to avoid unnecessary complications associated with tissue destruction [8]. Spontaneous disease regression has been reported to occur in approximately 30% of DT cases [9].

In 2012, in light of the available treatment guidelines that had shifted from the previous universal surgical approach for all patients to a more conservative approach centered on medical treatment or patient management through active surveillance [10], the policy of our institution changed accordingly. Active surveillance is generally the first approach for most people with a new diagnosis of DT and consists of active monitoring or active observation by check-up visits with the oncologist every three to four months. At each visit, the decision to stop active surveillance and start specific therapy is based on clinical and radiological examinations (such as ultrasound or magnetic resonance imaging), and may be driven by the worsening of symptoms that may compromise quality of life (e.g., pain that cannot be managed with medication); rapid increase in tumor size; tumor growth leading to the proximity or compression of important structures, such as organs, large vessels, or nerves; or continued progression over time. In such cases, a multidisciplinary team (including oncologists, surgeons, pathologists, radiologists, and pain therapists) with expertise in DT treatment evaluates both the clinical situation and patient characteristics to choose the best treatment for each patient.

This study aimed to evaluate event-free survival (where the event was either a recurrence after initial surgical treatment or a change in the therapeutic management) and pain management according to different treatment strategies.

## 2. Materials and Methods

We retrospectively evaluated all patients with a histologically confirmed DT who were treated at the University Hospital of Padua (UHP) and at the Veneto Institute of Oncology (IOV) between 2001 and 2021. The pathological diagnosis of patients who were referred from outside facilities was revised by two pathologists with expertise in soft tissue sarcomas (MS, APDT). Diagnostic and therapeutic decisions were taken by a multidisciplinary team, including surgeons, oncologists, radiotherapists, radiologists, pathologists, and palliative physicians. Follow-up imaging was carried out with magnetic resonance with contrast. The surgical margins were classified according to the International Union Against Cancer classification [11]. The response evaluation criteria in solid tumors (RECIST) criteria were used to assess disease progression and treatment response [12].

All data were extracted from a prospectively maintained electronic database. The data included patient demographics, tumor characteristics, and treatment information.

Therapeutic strategy was classified as surgical, pharmacological, or active surveillance. Given the shift in the institutional policy in 2012, we divided the patients into four subgroups: those treated surgically before 2012 (SGPre12), those treated surgically after 2012 (SGPost12), those treated pharmacologically after 2012 (MG), and those under active surveillance after 2012 (ASG).

Statistical analysis was carried out using R 4.1 (R Foundation for Statistical Computing, Vienna, Austria) [13]. Continuous data were summarized as the median and interquartile range (IQR). The data were compared among treatment groups using the Kruskal–Wallis test (continuous data) and Fisher’s test or the Chi square test (categorical data). Changes in the presence of painful symptoms after treatment were assessed using the McNemar test. Event-free survival (EFS) was calculated from the date of the diagnosis to the date of the event or the last follow-up visit, where the event was either (i) the recurrence after initial surgical treatment or (ii) a change in the therapeutic management after initial non-surgical treatment. Survival curves were calculated using the Kaplan–Meier method. All tests were 2-sided, and a *p*-value < 0.05 was considered statistically significant.

This study was approved by the Ethics Committee of the Veneto Institute of Oncology (CESC-IOV approval n°. 2014/91) and was conducted according to the Helsinki Declaration principles. All patients gave their consent to have their anonymized data used for scientific purposes.

## 3. Results

Among the 136 eligible DT patients (88 females and 48 males, median age of 42 years), 13 patients were excluded because of missing data or unclear reporting about their treatment. Hence, the analysis included 28 patients who underwent surgical treatment before 2012 (SGPre12), 41 patients who underwent surgical treatment after 2012 (SGPost12), 38 patients who received pharmacological treatment (MG), and 16 patients who were under active surveillance (ASG). The patient characteristics are summarized in Table 1. Pain at diagnosis and a visceral site were more common in patients receiving pharmacological treatment (*p* < 0.0001 and *p* = 0.03, respectively). These patients also had larger tumors compared to those in other groups (*p* = 0.001). Overall, 46 women had a pregnancy before DT diagnosis. 

The chemotherapy treatment was a combination of intravenous Methotrexate 30 mg/m^2^ and Vinorelbine 20 mg/m^2^, administered every 10 to 14 days. Adverse events in chemo-treated patients were mostly low-grade according to the Common Toxicity Criteria for Adverse Events (CTCAE), such as grade 1–2 leucopenia and anemia (62%), grade 1–2 increase in transaminase (28%) (with a grade 3 increase in ALT in one patient), nausea and vomiting (17%), and fatigue (12%).

One SGPost12 and three MG patients underwent follow-up visits elsewhere and their follow-up data could not be retrieved for the analysis. Pharmacological treatment (MG) resolved painful symptoms in 16/27 (60%) patients (*p* = 0.0001). Surgical treatment reduced the number of patients reporting painful symptoms from 7 to 2 (SGPre12) and from 11 to 4 (SGPost12) (both *p* = 0.07), while active surveillance did not affect painful symptoms (*p* = 0.99).

The median follow-up duration was 40 months (IQR 23–74). During follow-up, 13 (46%) SGPpre12 and 17 (43%) SGPost12 patients had a recurrence, which was associated with painful symptoms in 4 patients (2 in each group). The recurrence was treated with surgery (17 patients), chemotherapy (5 patients), NSAID/HT (3 patients), or active surveillance (5 patients).

Twenty-four (68%) MG patients underwent a therapeutic change due to disease progression (*n* = 7), unsatisfactory response with pain (*n* = 8), pain (*n* = 4), adverse effects of the treatment (*n* = 3), or partial disease response and/or pain resolution (*n* = 2). The therapeutic change was a shift in the medical treatment, except for three patients who interrupted the treatment due to adverse effects or disease progression, and the two patients with partial response and/or pain resolution who were put under active surveillance.

Nine (56%) ASG patients underwent a therapeutic change due to disease progression (*n* = 3), exacerbation of the pain (*n* = 3), or disease progression with exacerbation of the pain (*n* = 3). Systemic treatment included chemotherapy (two patients) or NSAID/HT (five patients). The other two patients continued their treatment at outside facilities and were lost at follow-up.

Event-free survival (where the event was recurrence after initial surgical treatment or a change in therapeutic management after initial non-surgical treatment) at 1, 3, and 5 years was: 85%, 70%, and 62% in the SGPre12 group; 76%, 58%, and 49% in the SGPost12 group; 49%, 31%, and 31% in the MG group; and 45%, 45%, and 45% in the ASG group (Figure 1).

Among the female subjects, event-free survival at 1, 3, and 5 years was 69%, 61%, and 57% in women with pregnancy before DT diagnosis, and 50%, 40%, and 40% in women who did not have pregnancies before DT diagnosis (*p* = 0.10).

## 4. Discussion

This study investigated event-free survival and pain management in DT patients undergoing different treatment strategies, including a surgical approach, medical therapy, and active surveillance. Overall, our findings show that event-free survival was higher in surgical groups (where the event was a recurrence of the disease), while non-surgical groups showed an early decline of the survival (where the event was a change in therapeutic management). Of course, this information should be balanced with the different clinical implications of the two events. In fact, many patients who were initially treated with surgery underwent a second surgical intervention due to the recurrence, while none of the patients who received medical therapy or were under active surveillance were treated with surgery when a therapeutic change was needed. Moreover, two patients receiving medical therapy were shifted to active surveillance. Among patients treated with systemic therapy or under active surveillance, most experienced a change in therapeutic management within the first 12 months. However, this is not discouraging and supports the non-surgical first approach in DT [3]. This is a selection “ex adiuvantibus” that can spare an unnecessary surgical intervention to a considerable number of patients. Of note, we cannot exclude that some imbalances in tumor site and size among the treatment groups may have contributed to the observed survival differences. Visceral localization was more common in patients who underwent surgery after 2012, while somatic localization was common in the other groups. In visceral settings, surgical resections might be considered in case of potentially serious or fatal complications (such as occlusion or volvulus), thus justifying such an approach. Large tumors were more common in the medical therapy group, which could be explained by the purpose of offering a less invasive therapy (compared to extensive surgical resection) in patients with a high rate of pain. In addition, the survival curves of DT patients who underwent surgery before or after 2012 displayed a difference that could be explained by the above-mentioned shift in patient management. This change toward a more conservative approach [12] has probably led to a different assignment of patients to treatment options, limiting surgery only to more complex cases. In other words, we believe that some patients who underwent surgery before 2012 had a clinically favorable status and could have benefited from a less invasive approach, which was offered to similar patients after 2012, thus leading to a divergence in the survival curves. In our series, medical therapy provided a symptomatic resolution in terms of pain in 60% of patients, and no pain was reported by asymptomatic patients after medical treatment. Indeed, medical therapy could provide symptom relief in many DT patients, though analgesic therapy alone (i.e., pregabalin) might be equally effective. DT should not discourage women from planning a pregnancy [14], but they should be closely followed-up to detect potential changes in the disease. Of note, pregnancy may change DT behavior, but it does not increase obstetric risk [15]. Even though DT is considered a benign disease, the patient’s experience may be quite distressful because pain is a frequent symptom, treatments options may include chemotherapy and/or surgery, and patients should undergo frequent follow-up visits. Psychological distress has been increasingly evaluated in oncological settings, and some efforts have also been made for DT patients, who may experience persistent high levels of emotional distress [16,17,18]. Unfortunately, the retrospective nature of our study precluded the investigation of such an aspect. Another limitation was the limited sample size, which did not allow for the assessment of subgroups. For example, the comparison of event-free survival between visceral and non-visceral sites would have provided interesting information to the reader, but the small number of patients with a visceral site in the subgroups of SGpre12, MG, and ASG did not allow for a meaning analysis of such data. Another interesting aspect would be function at diagnosis and at different time points during the follow-up. Unfortunately, such data were not available in the retrospective data collection, and prospective investigations are required to shed light on this aspect. Finally, we acknowledge some unbalanced characteristics among treatment groups, such as visceral/somatic location and tumor size. Yet, the heterogeneity of clinical signs, the presence or absence of pain, and the risk of intestinal obstruction in abdominal locations resulted in the heterogeneous treatment of patients. Larger series of patients and prospective studies are ongoing and may offer a better understanding of this rare disease. A recent phase III study showed high efficacy of oral γ-secretase inhibitor nirogacestat vs. placebo in DT patients who experienced progression, with a response rate of 41% and 8% respectively [19]. Of note, the literature offers few options for response evaluation in DT patients. In our series, the RECIST system was used to evaluate tumor response in treated DT patients, but we believe that the recent heterogeneity in the treatment strategy for DT highlights the need for a more appropriate tool for response evaluation. In addition, there is no standardized approach to evaluating pain in DTs. Patient-reported outcome measures are of utmost importance both to assess and prospectively monitor patient’s pain experience. The most common scale for pain assessment in clinical practice is the numeric rating scale (NRS), which was available in our hospital charts and was considered in the pain analysis. Although it is a simple and quick scale, the NRS might not be the best option to capture the multiple aspects of pain in DT patients, and clinicians might consider other scales, such as the Brief Pain Inventory—Short Form (BPI-SF) [20] or the Gounder–Desmoid Tumor Research Foundation Desmoid Symptom/Impact Scale (GODDESS), Desmoid Tumor Symptom Scale (DTSS), and the GODDESS Desmoid Tumor Impact Scale (DTIS) [21]. Further studies may also provide useful insights on this aspect.

## 5. Conclusions

This study investigated event-free survival and pain management in DT patients according to different treatment strategies, including a surgical approach, medical therapy, and active surveillance. In summary, event-free survival was higher in surgical groups and showed an early drop in non-surgical groups, but this information should be balanced with the different clinical implications of the two events, i.e., recurrence in surgical groups or therapeutic changes in non-surgical groups.

Our study supports the role of active surveillance as initial management, provided that about half of the patients did not experience any progression, while surgery or other treatment approaches (Appendix A) may be considered as first-line treatment for selected patients. In terms of pain relief, medical therapy provided symptom resolution in more than half the cases.

## Figures and Tables

**Figure 1 jpm-13-01653-f001:**
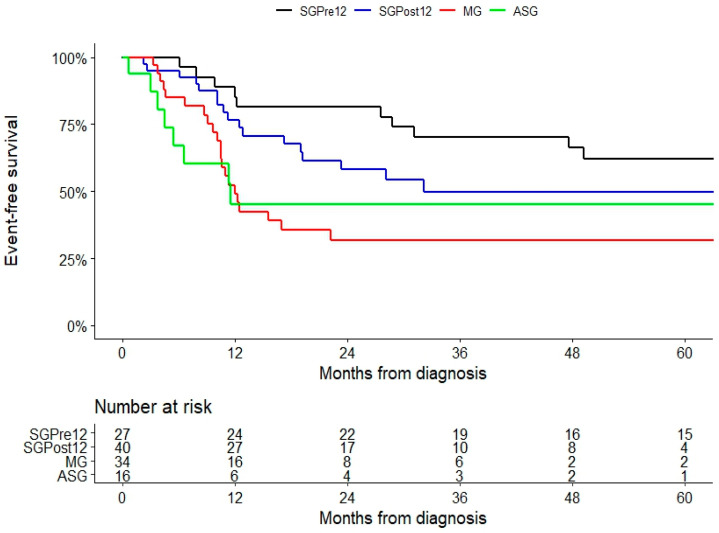
Event-free survival (where the event was recurrence after initial surgical treatment or a change in therapeutic management after initial non-surgical treatment) in patients with desmoid tumors who were treated at our institution between 2001 and 2021. ASG: patients under active surveillance; MG: patients receiving pharmacological treatment; SGPost12 patients who underwent surgical treatment after 2012; SGPre12 patients who underwent surgical treatment before 2012.

**Table 1 jpm-13-01653-t001:** Demographic, tumor, and treatment characteristics of patients with a histologically confirmed desmoid tumor who were treated at our institution between 2001 and 2021. Somatic sites included abdominal wall and extra-abdominal wall, while visceral sites included intra-abdominal wall. ASG: patients under active surveillance; CT: chemotherapy; HT: hormone therapy; IQR: interquartile range; MG: patients receiving pharmacological treatment; NSAID: non-steroidal anti-inflammatory drug; SGPost12 patients who underwent surgical treatment after 2012; SGPre12 patients who underwent surgical treatment before 2012; DT: desmoid tumor.

Variable	SGPre12 (*n* = 28)	SGPost12 (*n* = 41)	MG (*n* = 38)	ASG (*n* = 16)	*p*-Value
Males: *n* (%)	6 (21)	14 (34)	16 (42)	7 (44)	0.30
Age, years: median (IQR)	34 (27–48)	49 (36–56)	44 (32–60)	46 (36–55)	0.14
Dimension, mm: median (IQR)	40 (25–58)	50 (35–90)	79 (66–96)	48 (32–60)	0.001
Site: *n* (%) Somatic Visceral Multiple	26 (93) 2 (7) 0 (0)	26 (64) 14 (34) 1 (2)	33 (87) 5 (13) 0 (0)	14 (88) 2 (12) 0 (0)	0.03
Pain at diagnosis: *n* (%)	7 (25)	11 (26)	30 (79)	4 (25)	<0.0001
Pregnancy before DT diagnosis: *n* (%)	12/15 (80)	17/27 (63)	10/21 (48)	7/9 (78)	0.20
Type of diagnosis: *n* (%) Tru-cut biopsy Excisional biopsy Incisional biopsy Not reported	6 (21) 19 (68) 2 (7) 1 (4)	6 (15) 35 (85) 0 (0) 0 (0)	29 (76) 1 (3) 6 (16) 2 (5)	13 (81) 1 (6) 2 (13) 0 (0)	-
Surgical margins: *n* (%) R0 R1 R2 Not reported	6 (22) 16 (57) 2 (7) 4 (14)	18 (44) 16 (39) 1 (2) 6 (15)	-	-	-
Medical treatment: *n* (%) CT NSAID NSAID + HT HT	-	-	13 (34) 18 (48) 2 (5) 5 (13)	-	-
Medical treatment: *n* (%) CT NSAID NSAID + HT HT	-	-	13 (34) 18 (48) 2 (5) 5 (13)	-	-

## Data Availability

The data presented in this study are openly available in Zenodo at https://doi.org/10.5281/zenodo.10203542.

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
