# Peer review of "Desmoid Tumors in the Active Surveillance Era: Evaluation of Treatment Options and Pain Relief in a Single-Center Retrospective Analysis"

_jpm, 2023, doi:10.3390/jpm13121653_

Round 1

Reviewer 1 Report

Comments and Suggestions for Authors

Desmoid tumours are rare soft tissue tumours with aggressive infiltrative growth and local relapse in some cases. According to international guidelines, active surveillance is currently considered the first line of treatment for most DT patients. When active treatment is required, several systemic and local methods are considered. Recently, systemic therapy options include hormonal agents, nonsteroidal anti-inflammatory drugs, tyrosine kinase inhibitors, and anthracycline-based regimens. In this context, the manuscript prepared by

Marco Rastrelli et al. provide significant data regarding event-free survival and pain management according to different treatment strategies.

To improve the quality of the manuscript, I suggest only a few minor recommendations to the authors:

-     Kindly add a reference in line 56 according to the data presented.

-     Please revise the table 1 legend and explain all the abbreviations used in this table (e.g. DT)

-     Kindly revise the reference list according to the Journal of Personalized Medicine recommendations

-     Please add the Approval No. (from the Veneto Oncological Institute)in the Material and methods or the Institutional Review Board Statement.

Author Response

We thank the Reviewer for his/her comments, which improved the quality of the manuscript. You can find our answers below and please see the attachment of revised versions of the manuscript.

  • Kindly add a reference in line 56 according to the data presented .

Re:  done as requested

  • Please revise the table 1 legend and explain all the abbreviations used in this table (e.g. DT)

Re: done as requested

  •     Kindly revise the reference list according to the Journal of Personalized Medicine recommendations .

Re: done as requested

  •      Please add the Approval No. (from the Veneto Oncological Institute) in the Material and methods or the Institutional Review Board Statement. (Paolo: done as requested)

Re : done as requested)

Reviewer 2 Report

Comments and Suggestions for Authors

The authors of manuscript titled “Desmoid tumors in the active surveillance era: evaluation of treatment options and pain relief in a single-centre retrospective analysis” tried to evaluate event free survival after initial surgical treatment or change in therapeutic management after initial non-surgical treatment and pain management”

Authors need to address some issues

1-     Abstract: it is not clear what the target of the study in the abstract is. Authors gave information of the methods and results but did not address the problem clearly.

I advise the authors to re-write the authors and stated clearly the target of the study

2-     Introduction is good but it is short, authors need to extend the introduction

3-     Methods and results are represented good

4-     Discussing is explained well

5-     The conclusion is short and I advise the authors to summarize the study and the results in the conclusion

Author Response

Thank you for pointing out these criticisms, you can find our answers below and please see the attachment of revised versions of the manuscript.

  • Abstract: it is not clear what the target of the study in the abstract is. Authors gave information of the methods and results but did not address the problem clearly.I advise the authors to re-write the authors and stated clearly the target of the study

Re: done as requested; This study aimed to evaluate event-free survival and pain management according to different treatment strategies

  • Introduction is good but it is short, authors need to extend the introduction.

Re: done as requested; The initial diagnosis is based on imaging techniques (computed tomography and magnetic resonance imaging), which reveal the presence of an expanding infiltrating mass. The diagnosis is confirmed by biopsy of the tumour, which shows abundant collagen around elongated fusiform cells containing small, regular nuclei and pale cytoplasm. Immunohistological examination shows the expression of muscle cell markers (actin, desmin, vimentin) and the absence of CD34. In addition, the diagnosis can be confirmed by screening for CTNNB1 mutations. The differential diagnosis is wide ranging and includes, on the one, fibrosarcomas and, on the other, myofibroblastic processes such as nodular fasciitis, hypertrophic scars and keloids. The differential diagnosis of intra-abdominal DTs arises with gastrointestinal stromal tumours, isolated fibrous tumours, inflammatory myofibroblastic tumours, sclerosing mesenteritis and retroperitoneal fibrosis.

  • Methods and results are represented good.

Re: Thanks!

  • Discussing is explained well

Re: Thanks

  • The conclusion is short and I advise the authors to summarize the study and the results in the conclusion.

Re: done as requested; This study investigated event-free survival and pain management in DT patients according to different treatment strategies including surgical approach, medical therapy and active surveillance. In summary, event-free survival was higher in surgical groups while showed an early drop in non-surgical groups, but this information should be balanced by the different clinical implications of the two events, i.e. recurrence in surgical groups or therapeutic change in non-surgical groups.